# A Novel OFDM Format and a Machine Learning Based Dimming Control for LiFi

Itisha Nowrin [1], M. Rubaiyat Hossain Mondal [1], Rashed Islam [1] and Joarder Kamruzzaman [2,*]

1   Institute of Information and Communication Technology, Bangladesh University of Engineering and Technology (BUET), Dhaka 1000, Bangladesh; nowrin.turna@gmail.com (I.N.); rubaiyat97@iict.buet.ac.bd (M.R.H.M.); rashedece@gmail.com (R.I.)
2   School of Engineering, Information Technology and Physical Sciences, Federation University, Gippsland Campus, Churchill, VIC 3842, Australia
*   Correspondence: joarder.kamruzzaman@federation.edu.au; Tel.: +613-5122-6665

**Abstract:** This paper proposes a new hybrid orthogonal frequency division multiplexing (OFDM) form termed as DC-biased pulse amplitude modulated optical OFDM (DPO-OFDM) by combining the ideas of the existing DC-biased optical OFDM (DCO-OFDM) and pulse amplitude modulated discrete multitone (PAM-DMT). The analysis indicates that the required DC-bias for DPO-OFDM-based light fidelity (LiFi) depends on the dimming level and the components of the DPO-OFDM. The bit error rate (BER) performance and dimming flexibility of the DPO-OFDM and existing OFDM schemes are evaluated using MATLAB tools. The results show that the proposed DPO-OFDM is power efficient and has a wide dimming range. Furthermore, a switching algorithm is introduced for LiFi, where the individual components of the hybrid OFDM are switched according to a target dimming level. Next, machine learning algorithms are used for the first time to find the appropriate proportions of the hybrid OFDM components. It is shown that polynomial regression of degree 4 can reliably predict the constellation size of the DCO-OFDM component of DPO-OFDM for a given constellation size of PAM-DMT. With the component switching and the machine learning algorithms, DPO-OFDM-based LiFi is power efficient at a wide dimming range.

**Keywords:** machine learning; orthogonal frequency division multiplexing; dimming; light fidelity; regression





## 1. Introduction

The demand for high bandwidth is increasing rapidly, resulting in a spectrum crunch. As a result, there is a need for higher frequency for wireless application scenarios. The available radio spectrum below 10 GHz is insufficient, and this has prompted new research on wireless communication to increase the bandwidth. Terrestrial communication at a very high frequency can be an option; however, the infrastructure is the biggest challenge [1]. Light fidelity (LiFi) is the next generation solution for wireless communication where light-emitting diodes (LEDs) are used. In LiFi, LEDs transmit data using intensity modulation (IM), and the receiver receives data using direct detection (DD). Though LiFi and visible light communication (VLC) are both considered as wireless communication using LEDs with digital modulation, there are significant differences between them. VLC is considered as a point-to-point connection, whereas LiFi is a point to multipoint and multipoint-to-point communication enabling full user mobility [1–3]. Optical wireless communication (OWC) has earned much popularity as a technology that complements radio frequency (RF) communication. Since LiFi is a form of OWC technology, it uses high-speed LEDs to provide data in a full network along with room illumination. Multiple designs for handover modeling are reported in the context of LiFi [4–6]. LiFi technology needs a modulation technique to carry information in dimmable illumination environments [7–11]. Some popular single carrier modulation (SCM) schemes for LiFi are on–off keying (OOK),

pulse position modulation (PPM), and pulse amplitude modulation (PAM). However, these schemes suffer from various unwanted effects such as nonlinear signal distortion with the increase in data rates. A complex equalization process is required for overcoming such effects. To avoid this complex equalizer, as an alternative solution, multi-carrier modulation (MCM) is also applied. The most used technique of MCM in LiFi is orthogonal frequency division multiplexing (OFDM). OFDM provides high data rates and prevents inter symbol interference (ISI) by using a single tap equalizer at the receiver. Since light intensity cannot be negative, optical OFDM is unipolar. To obtain unipolar OFDM without any loss of information, there are several derivatives of OFDM, including asymmetrically clipped optical OFDM (ACO-OFDM) [12,13], DC biased optical OFDM (DCO-OFDM) [12,14–16], and pulse amplitude modulated discrete multitone modulation (PAM-DMT) [17,18]. In addition, there are some hybrid forms of OFDM such as asymmetrically clipped DC-biased optical OFDM (ADO-OFDM) [12,16], hybrid asymmetrically clipped optical OFDM (HNC-OFDM) [16], hybrid asymmetrically clipped optical OFDM (HACO-OFDM) [19], hybrid diversity combined OFDM (HDC-OFDM) [16,20], asymmetrical hybrid optical OFDM (AHO) [21], hybrid DC biased asymmetrically clipped pulse amplitude modulated discrete multitone modulation optical OFDM (HDAP-OFDM) [22], and pulse amplitude-modulated hybrid optical OFDM (PHO-OFDM) [23]. In the rest of the paper, the terms ACO-OFDM, DCO-OFDM, AHO-OFDM, HACO-OFDM, HDAP-OFDM, and PHO-OFDM are simplified as ACO, DCO, AHO, HACO, HDAP, and PHO, respectively.

In PHO, M-PAM and M-QAM were used on odd subcarriers and even subcarriers, respectively [23]. However, the dimming range of PHO-based LiFi was not studied in [23]. In HDAP, DCO was used in the lower subcarriers [22]. On the other hand, ACO was used in the higher odd subcarriers, and PAM-DMT was used in the higher even subcarriers of HDAP. The overall dimming range for HDAP was studied, but the required proportion of the elements of HDAP for a given dimming target has not been investigated before [22]. In this research, a switching algorithm for HDAP is proposed where the individual components of HDAP are switched according to a target illumination level. This paper proposes a new hybrid modulation approach termed DC-biased pulse amplitude modulated optical (DPO-OFDM), where DCO will be kept in the odd subcarriers, and PAM-DMT will be kept in the imaginary portion of the even subcarriers. It is shown later in this paper that the proposed DPO-OFDM, simplified as DPO, can achieve a wide dimming range with better spectral efficiency compared to ADO and has better power efficiency compared to AHO and HDAP. It can be noted that finding the appropriate proportions of the individual components of DPO, HDAP, or any other hybrid OFDM is an important task where machine learning (ML) may be useful. This paper also focuses on the effectiveness of applying ML in selecting the components of hybrid OFDM formats.

The major contributions of this article are in the following aspects.

1.  A new OFDM format termed DPO-OFDM is introduced using the concepts of DCO-OFDM and PAM-DMT;
2.  Mathematical expressions of the DC-bias level for the proposed DPO-OFDM are derived. The BER performance and spectral efficiency with dimming control are compared with existing optical OFDM formats;
3.  A switching algorithm for the existing HDAP-OFDM is proposed where the individual components of HDAP-OFDM are switched according to a target dimming level;
4.  ML regressors are used to find the appropriate proportions of the individual components of DPO-OFDM for a target dimming level. This concept is also applicable to HDAP-OFDM and other hybrid OFDM formats.

The rest of the paper is arranged as follows: Section 2 reviews the literature of existing OFDM schemes. Section 3 describes the proposed DPO system. The framework of the DC-bias level of the DPO system is presented in Section 4. Section 5 provides performance results of DPO in terms of power efficiency and dimming control. In Section 6, a switching algorithm for HDAP is presented. Section 7 describes an ML-based approach for finding

the appropriate proportions of the elements of DPO or other hybrid OFDM schemes for a given dimming level. Finally, Section 8 presents the concluding remarks.

## 2. Literature Review on OFDM Formats

This section provides a review of the existing notable works on OFDM-based LiFi. This review is not performed using any evidence-based systematic review method; instead, it is performed by searching articles from reputed databases. Several keywords, including LiFi, visible light communication (VLC), optical OFDM, machine learning, and dimming, were used to search articles from IEEE, Elsevier, The Optical Society, and Springer repositories.

LiFi uses infrared and visible light spectra to provide multiuser access and user mobility. In this section, the strategies of OFDM techniques used in the LiFi system are briefly studied. With an equalizer at the receiver, OFDM techniques provide a high data rate and prevent inter-symbol interference (ISI), making OFDM more attractive for LiFi [23–29] technology. OFDM was widely applied in the context of wireless radio channels to combat ISI and to provide high data rates [30,31]. OFDM also has the potential to be useful for LiFi. Some of the popular unipolar OFDM techniques suitable for LiFi are DCO, ACO, and PAM-DMT. In DCO, both odd and even subcarriers carry data to transmit information. A DC bias is added to the bipolar signal. After that, the remaining negative part of the signal is clipped. This clipping generates a clipping noise in the DCO signal. In ACO, only odd subcarriers carry data to transmit information, and there is no additional DC bias such as DCO. The signal is clipped at zero to make it unipolar, and the clipping noise falls in the even part. In the case of power efficiency, DCO is less efficient than ACO for small constellation sizes, but for large constellations, ACO becomes inefficient. On the other hand, DCO exhibits better spectral efficiency than ACO since DCO uses both odd and even subcarriers, while ACO uses only odd subcarriers. PAM-DMT is another optical OFDM technique and is suitable for IM/DD [18]. In PAM-DMT, the pulse amplitude modulation of the subcarriers is used. In the conventional PAM-DMT, only the imaginary parts of the subcarriers carry data, and the real parts experience clipping noise. Both PAM-DMT and ACO use asymmetrical clipping and show better power efficiency compared to DCO.

In ADO, a combination of ACO and DCO is used. Odd subcarriers carry ACO data, and even subcarriers carry DCO data. ADO has better spectral efficiency compared to ACO and better optical power efficiency compared to DCO. In [27], a different combination of ACO and DCO was proposed, called asymmetrical and direct current biased OFDM (AAD-OFDM). In AAD-OFDM, the spectral efficiency is improved with a wide dimming range using ACO and DCO. According to the required dimming level, the scheme of encoding is altered between ACO and DCO. Thus, the performance is improved, and the system fully utilizes the dynamic range at any dimming level. In AAD-OFDM, ACO performs well in low and high brightness levels due to half-wave symmetry. Spectral efficiency is also greater in both conditions with full dynamic range. On the other hand, DCO occupies all the subcarriers, and the spectral efficiency is also greater in the mid dimming level. Therefore, in the AAD-OFDM system, when mid dimming is required, the system is switched to DCO, and for high and low dimming requirements, it is switched to ACO [24].

The study in [28] introduces pre-distorted enhanced ADO (PEADO-OFDM), where the receiver is designed in such a way that takes care of the latency and the complexity issues. In HACO [19], a combination of ACO and PAM-DMT is used. In this system, ACO and PAM-DMT are transmitted together by sending ACO in odd subcarriers and PAM-DMT even in imaginary subcarriers. No DC bias is needed in this scheme. The clipping noise is calculated at the receiver and eliminated to retrieve PAM-DMT symbols. The advantage of this scheme is the increased data rate. Moreover, no additional DC bias is used, and the spectral efficiency is improved. There is a slight degradation in signal-to-noise ratio (SNR), but there is an improvement in the peak to average power ratio (PAPR). The efficiency of the HACO system is improved in [29], where PAPR is reduced compared to ACO. This system results in a low BER of $10^{-7}$. AHO is a combination of ACO and PAM-DMT [21].

In AHO, the ACO signal is added to the inverted PAM-DMT signal. This scheme provides a wide dimming range with better spectral efficiency than HACO [21]. In HDAP, DCO symbols are placed in the lower subcarriers, while ACO and PAM-DMT symbols are placed in the higher odd and higher even subcarriers, respectively. This HDAP scheme is optically power efficient with a low PAPR and provides a wide dimming range.

### 3. System Description of DPO

This section describes a DPO-based LiFi scheme.

#### 3.1. DPO Transmitter

The transmitter block is described in Figure 1, where binary bits of DCO and PAM-DMT are given as input individually. Initially, the binary data for DCO are mapped to a QAM modulator, and the binary data for PAM-DMT are mapped to a PAM-modulator; both are then converted from serial to parallel. The input vector maintains Hermitian symmetry. For this property, and for the unused zeroth (DC) subcarrier, independent data are carried by only $N/2$-1 out of a total of $N$ subcarriers. Bits of DCO are mapped to odd subcarriers, and it carries $N/4$ independent data without Hermitian symmetry. Now the input vector of the IFFT block for DCO is as follows.

$$X_{DCO} = \left[0, X_1, 0, X_2, 0, \ldots, X_{N/2-1}, 0, X^*_{N/2-1}, 0, \ldots, X^*_0\right] \tag{1}$$

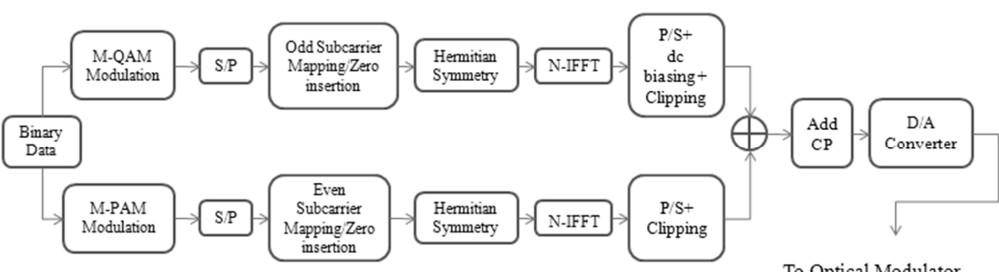

**Figure 1.** DC-biased pulse amplitude modulated orthogonal frequency division multiplexing (DPO-OFDM) transmission side where dc-biased optical OFDM (DCO-OFDM) and pulse amplitude modulated discrete multitone (PAM-DMT) electrical signals are formed and then combined and converted to the optical domain. Note that QAM stands for quadrature amplitude modulation; while IFFT, S/P, P/S, CP and D/A stand for inverse fast Fourier transform, serial-to-parallel, parallel-to-serial, cyclic prefix, and digital-to-analog, respectively.

PAM-DMT modulates the imaginary part of even subcarriers. The input of IFFT for PAM-DMT can be expressed as

$$Y_{PAM} = \left[0, 0, Y_0, 0, Y_1, 0, \ldots, Y_{N/2-2}, 0, 0, 0, Y^*_{N/2-2}, 0, \ldots, Y^*_0, 0\right] \tag{2}$$

The outputs of both IFFT blocks are time domain signals. Next, a DC bias, BDC, is added to the DCO time domain signal. The biased signal is then clipped at amplitude zero. The time domain PAM-DMT parallel signal is converted to serial. Next, the serial signal is clipped to zero and specified by $x_{DCO,C}$ and $y_{PAM,C}$ for DCO and PAM-DMT, respectively. After clipping, these signals are added together, and then a cyclic prefix is added. The resultant signal is denoted by $z$, where $z_w$ is the wanted signal and $n_{DCO}$ and $n_{PAM}$ are the clipping noise of DCO and PAM-DMT, respectively.

$$z = (x_{DCO,C} + B_{DC}) + y_{PAM,C} \tag{3}$$

The expression in (3) can be rearranged as follows.

$$z = x_{DCO,C} + B_{DC} + y_{PAM,C} \tag{4}$$

The signal $z$ in (4) can be expressed in terms of the wanted signal and clipping noise in the following format.

$$z = z_w + B_{DC} + n_{DCO} + n_{PAM} \tag{5}$$

The combined DPO signal is then transformed to the analog domain. This is performed with the use of a digital-to-analog (D/A) converter. Next, optical modulators, for example, LEDs, are used to transform the analog signal from the electrical to the optical domain.

### 3.2. DPO Receiver

The DPO receiver is presented in Figure 2. In the receiver, a reverse operation is performed. A photodetector detects the transmitted signal, and the signal is converted to the electrical domain, which is specified by $\hat{z}$ as follows.

$$\hat{z} = z + n_{AWGN} \tag{6}$$

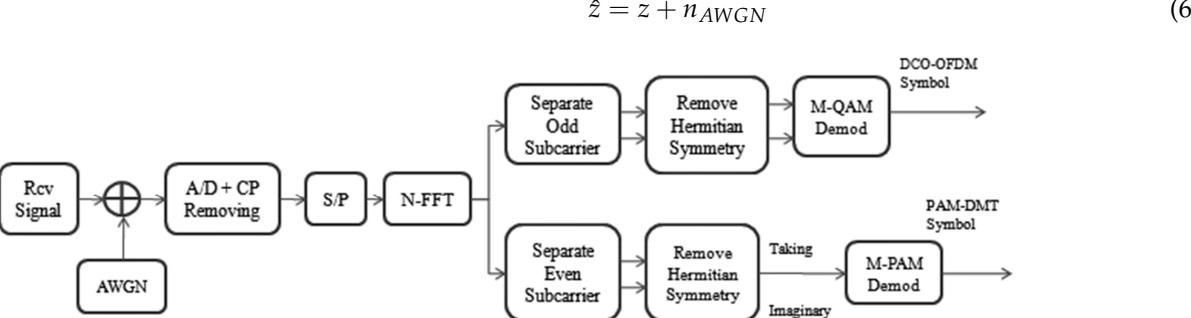

**Figure 2.** DC-biased pulse amplitude modulated orthogonal frequency division multiplexing (DPO-OFDM) receiver transforms the optical signal into electrical and then separates the dc-biased optical OFDM (DCO-OFDM) and pulse amplitude modulated discrete multitone (PAM-DMT) components. Note that Rcv stands for received, while A/D, CP, S/P and IFFT stand for analog-to-digital, cyclic prefix, serial-to-parallel, and inverse fast Fourier transform.

In (6), $n_{AWGN}$ is the additive white Gaussian noise (AWGN). The received elements are $\hat{x}_{DCO}$ for the transmitted DCO signal elements $x_{DCO}$, and $\hat{y}_{PAM}$ for the transmitted PAM-DMT signal element. The combined received signal is converted from analog to digital by an A/D converter, and the cyclic prefix is removed and converted from serial to parallel. Finally, it takes the following form:

$$\hat{z} = \hat{x}_{DCO} + \hat{y}_{PAM} + n_{DCO} + n_{PAM} + n_{AWGN} \tag{7}$$

The signal is then sent to a fast Fourier transform (FFT) operator to generate frequency domain signal, $\hat{Z}$.

$$\hat{Z} = \hat{X}_{DCO} + \hat{Y}_{PAM} + N_{DCO} + N_{PAM} + N_{AWGN} \tag{8}$$

On the receiver side, photodetectors convert the incoming optical signal to the electrical domain. The DCO symbols carried by the odd subcarriers are recovered first. After removing Hermitian symmetry, it is demodulated using an M-QAM demodulator. Note that the DCO signal may contain clipping noise, but it can be negligible only by applying an appropriate DC bias. In the same process, PAM-DMT is recovered using an M-PAM demodulator. It can be noted that noise due to clipping in PAM-DMT affects each even subcarrier on its real part. Therefore, it does not have any effect on the signal. Finally, the even parts of imaginary subcarriers are extracted.

### 4. Analysis of DC Bias for DPO

In this section, we analyze the DC bias termed as $I_{Bias}$ for DPO. This is an extension of the concepts that are described in [5,15]. Here, $I_{Bias}$ is defined as the bias to make unipolar signals from the composite signals. Now, if the root mean square (RMS) of the unclipped

DCO and PAM-DMT are labeled as $\sigma_{DCO}$ and $\sigma_{PAM}$, respectively, then we can write the average amplitude of the DPO signal as

$$I_D = \frac{\sigma_{DCO}}{\sqrt{2\pi}} + \frac{\sigma_{PAM}}{\sqrt{2\pi}} + I_{Bias} \tag{9}$$

The scaling factor of DCO is expressed as $\beta_{DCO} = \frac{I_H - I_{Bias}}{\sigma_{DCO}}$. Similarly, the scaling factor of PAM-DMT is $\beta_{PAM} = \frac{I_H - I_{Bias}}{\sigma_{DCO}}$. In both cases, $I_H$ and $I_L$ are the maximum value and the minimum value of the signal constrained by LED. Furthermore, the dimming level, $\eta$, is defined by,

$$\eta = \frac{I_D - I_{Bias}}{I_H - I_L} \tag{10}$$

The optical intensity or brightness will be higher depending on the higher dimming level. The expression (10) can be written as

$$I_D = I_L(1 - \eta) + \eta I_H \tag{11}$$

By rearranging (9), we obtain the following expressions:

$$I_{Bias} = I_D - \frac{\sigma_{DCO}}{\sqrt{2\pi}} - \frac{\sigma_{PAM}}{\sqrt{2\pi}} \tag{12}$$

$$I_{Bias} = I_D - \frac{I_H - I_{Bias}}{\sqrt{2\pi}\beta_{DCO}} - \frac{I_H - I_{Bias}}{\sqrt{2\pi}\beta_{PAM}} \tag{13}$$

Using the value of root mean square as $\sigma_{DCO}$ and $\sigma_{PAM}$, we can finally express $I_{Bias}$ as,

$$I_{Bias} = \frac{\{I_L - \eta(I_L - I_H)\}\sqrt{2\pi}\beta_{DCO}\beta_{PAM} - I_H\beta_{DCO} - I_H\beta_{PAM}}{\sqrt{2\pi}\beta_{DCO}\beta_{PAM} - \beta_{DCO} - \beta_{PAM}} \tag{14}$$

Therefore, $I_{Bias}$ depends on the scaling factor of the individual components of the technique and $I_L$ and $I_H$. It is a complex function, and by varying the value of $I_{Bias}$, different dimming levels are obtained.

The dimming levels versus bias values are plotted in Figure 3 using (14) with $I_H = 1$ and $I_L = 0$, following the concept of [5,15]. It illustrates the plots of $I_{Bias}$ versus $\eta$ for different scaling factors. It is observed that with the increase in the scaling factor, a wider dimming range can be obtained. It can be shown that a wide dimming range can be obtained for the case of larger values of a scaling factor. Since the level of the average amplitude of the combined signal is higher than the DC bias, with the increment of DC bias, the dimming level increases. The combined signal suffers slightly at low and high dimming levels due to DCO element clipping.

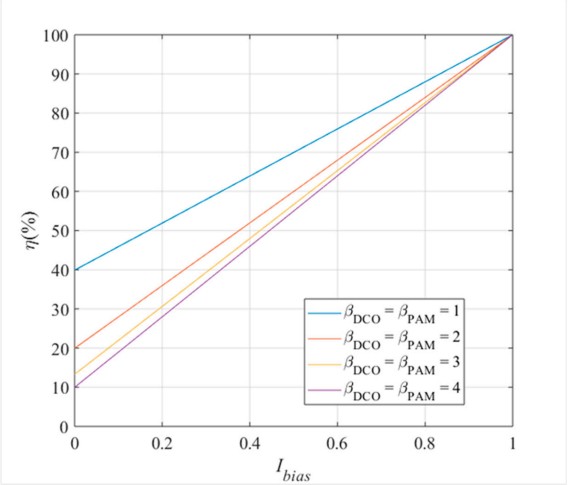

**Figure 3.** Dimming level, $\eta$, versus bias, $I_{Bias}$, values in dc-biased pulse amplitude modulated orthogonal frequency division multiplexing (DPO-OFDM), where $\beta$ terms represent scaling factors.

## 5. Performance Evaluation of DPO

The performance results of DPO and other OFDM schemes are obtained here with the help of simulations with the MATLAB (version 9.2) tool. The simulations were performed for an AWGN channel without considering any other sort of distortion. In order to keep consistency with the results described in the literature [12,22], the target uncoded BER was considered to be $10^{-3}$.

This section discusses the power efficiency of the proposed LiFi system. The performance is evaluated using two terms. One is the electrical energy per bit to noise power spectral density, $E_{b(elec)}/N_O$. In this case, the bit rate/normalized bandwidth (bit/s/Hz) is expressed as $R$. The other one is the optical energy per bit to noise power spectral density, $E_{b(opt)}/N_0$. Figure 4 shows the BER versus $E_{b(elec)}/N_O$ when $R$ is 2. It can be seen that both ADO and DPO have excellent efficiency in terms of $E_{b(elec)}/N_O$. Figure 5 presents $E_{b(elec)}/N_O$ versus $R$ for the proposed DPO system. For comparison, plots are also provided for HDAP (50% DCO and 50% ACO-PAM), AHO, ADO, and DCO. Here we can see that for $R$ of 2, 3, and 4, DPO requires a lower $E_{b(elec)}/N_O$ than HDAP and AHO. For example, at an R of 3 for a BER $10^{-3}$, DPO requires exactly 2 dB and 7 dB less $E_{b(elec)}/N_O$ than HDAP (50% DCO and 50% ACO-PAM) and AHO, respectively. Compared to DCO, DPO has 2 dB and 3 dB more electrical power efficiency at $R$ of 2 and 4, respectively. However, DPO has the same electrical power efficiency as DCO at an $R$ of 3, and DPO is less efficient than ADO in terms of $E_{b(elec)}/N_O$. Note that in DPO, an $R$ of 2 is achieved using 4-QAM DCO and 4-PAM-DMT, R of 3 is achieved using 8-QAM DCO and 8-PAM-DMT, and $R$ of 4 is obtained with 16-QAM DCO and 16-PAM-DMT. Figure 5 also indicates that the required $E_{b(elec)}/N_O$ increases for an increase in the value of $R$. For example, for a given BER of $10^{-3}$, DPO requires $E_{b(elec)}/N_O$ of 16 dB, 24 dB, and 25 dB when $R$ has values of 2, 3, and 4, respectively.

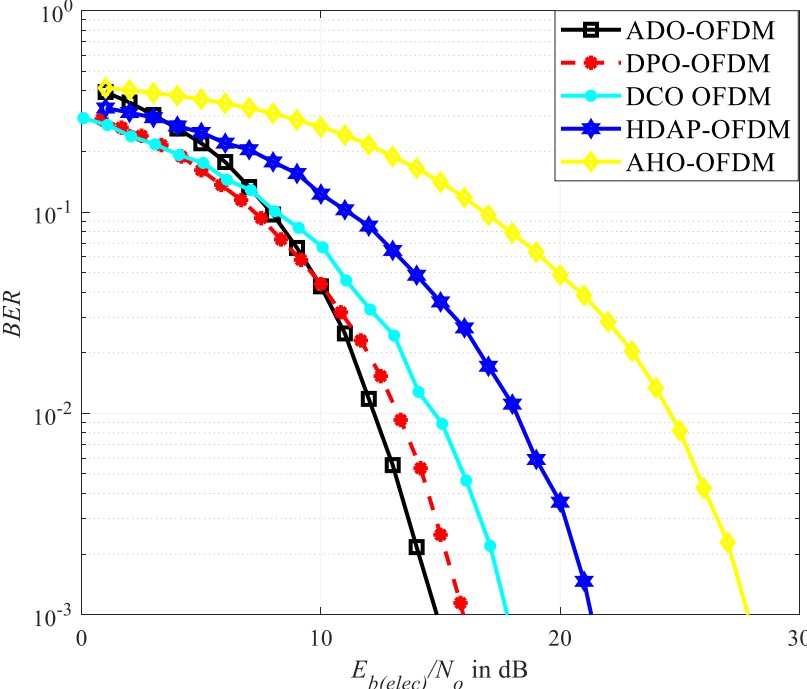

**Figure 4.** Bit error rate (BER) versus electrical energy per bit to noise power spectral density ($E_{b(elec)}/N_O$) for different orthogonal frequency division multiplexing (OFDM) forms.

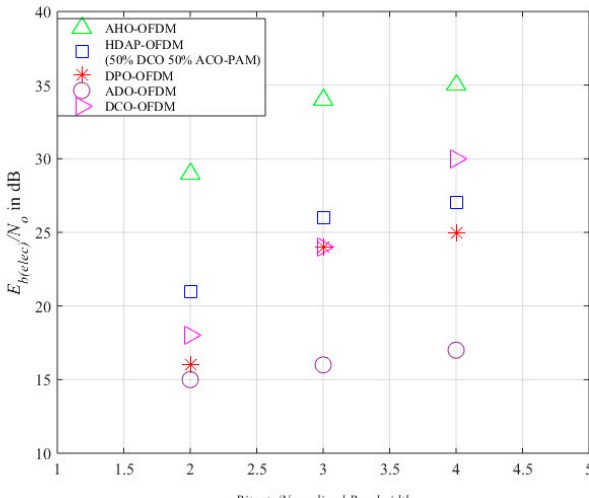

**Figure 5.** Electrical energy per bit to noise power spectral density ($E_{b(elec)}/N_O$) versus bit rate/normalized bandwidth ($R$) for hybrid orthogonal frequency division multiplexing (OFDM) considering a bit error rate (BER) of $10^{-3}$.

Figure 6 shows the BER versus $E_{b(opt)}/N_0$ when $R$ is 2. It can be seen that ADO has the best $E_{b(opt)}/N_0$ efficiency followed by DCO and DPO. Figure 7 demonstrates $E_{b(opt)}/N_0$ versus $R$ for DPO, HDAP (50% DCO and 50% ACO-PAM), AHO, ADO, and DCO. Here we can see that for $R$ of 2, 3, and 4, DPO requires a lower $E_{b(opt)}/N_0$ than AHO, but higher than ADO. For $R$ of 2, DPO requires approximately 4 dB lower $E_{b(opt)}/N_0$ than HDAP, but for $R$ of 3 and 4, DPO requires slightly less $E_{b(opt)}/N_0$ than HDAP. DPO is more optically power-efficient than DCO at an $R$ of 4. Next, simulations were performed to investigate the effect of the number of OFDM subcarriers on the BER performance for different OFDM forms in an AWGN channel. The total number of OFDM subcarriers, $N$, was varied from 64 to 1024 as these are the numbers reported for OFDM-based LiFi [12,19,26]. In the simulations, the total transmitted optical power was kept fixed for different values of $N$. Figure 8 shows the BER versus $E_{b(elec)}/N_O$ for an $R$ of 2 in an AWGN channel. For clarity of the illustration, the plots are shown only for DPO and HDAP, with $N$ having only three values: 128, 512, and 1024. Figure 8 presents that for a given transmitted optical power and for AWGN, the BER performance does not change with the change in $N$.

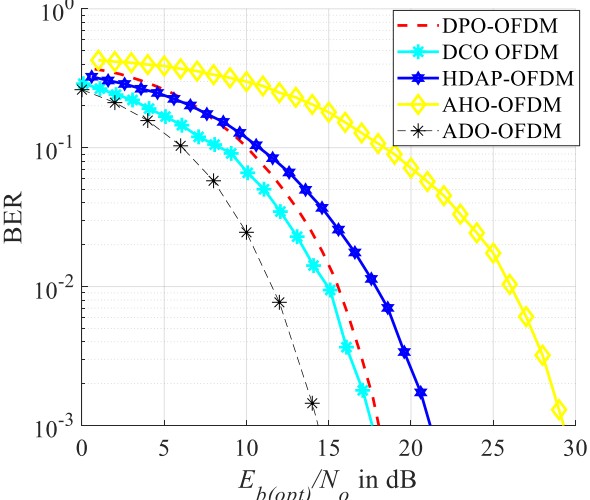

**Figure 6.** Bit error rate (BER) versus optical energy per bit to noise power spectral density ($E_{b(opt)}/N_0$) for different orthogonal frequency division multiplexing (OFDM) forms.

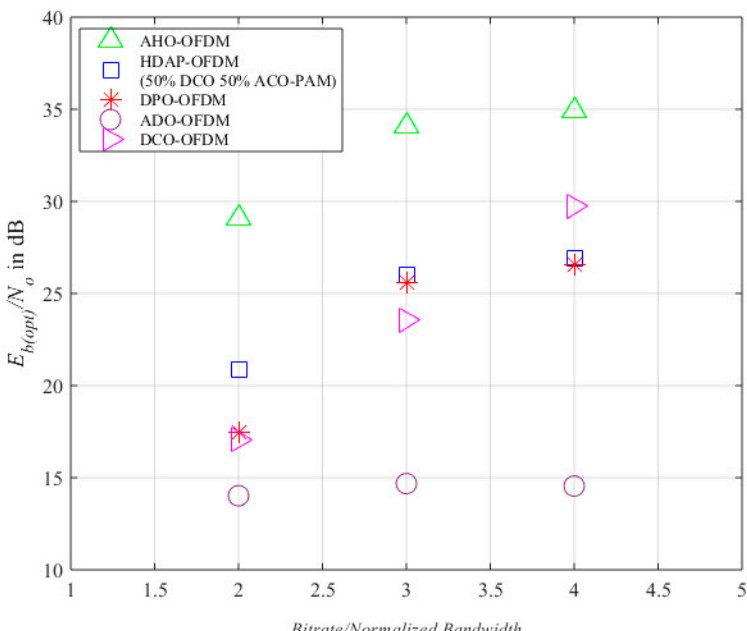

**Figure 7.** Optical energy per bit to noise power spectral density ($E_{b(opt)}/N_0$) versus bit rate/normalized bandwidth ($R$) for different orthogonal frequency division multiplexing (OFDM) forms considering a bit error rate (BER) of $10^{-3}$.

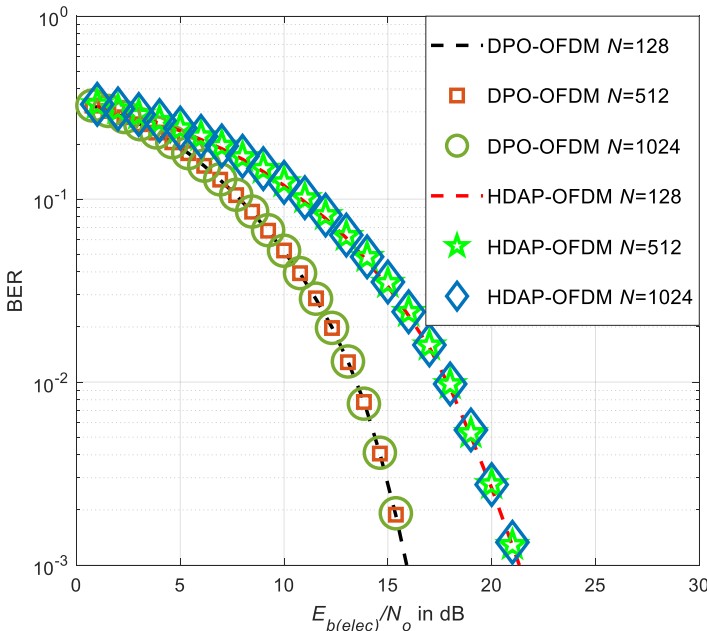

**Figure 8.** Bit error rate (BER) versus electrical energy per bit to noise power spectral density ($E_{b(elec)}/N_O$) for dc-biased pulse amplitude modulated orthogonal frequency division multiplexing (DPO-OFDM) and hybrid DC biased asymmetrically clipped pulse amplitude modulated discrete multitone modulation optical OFDM (HDAP-OFDM) for $N$ subcarriers in a additive white Gaussian noise channel.

## 6. Dimming Control for DPO and HDAP Systems

In this section, the dimming flexibility of different hybrid OFDM forms is discussed. Firstly, dimming control is discussed for DPO, HDAP, and others. Secondly, the dimming control of HDAP is discussed by switching the individual components.

For dimming control of a LiFi transmitter, simulations using MATLAB were performed. For this, the concepts reported in [22,27] were used. Figure 9 shows the performance of the

dimming range with respect to R of DPO, HDAP (50% DCO and 50% ACO-PAM), AHO, DCO, and ADO. For dimming flexibility, the size of constellation points is changed so that different plots are obtained. For instance, to attain a 50% dimming, 16-PAM and 4-QAM are used for PAM-DMT and DCO elements, respectively. Note that for the plots, the target BER is considered to be $10^{-3}$. The achievable dimming range for DPO is 3–97%, which is similar to HDAP. The dimming range of both AHO and DCO is 2–98%, and that of ADO is 5–96%. For a target dimming value, all OFDM forms will carry the same optical power. This means, at a certain diming level, if the scheme contains better *R*, it will have better optical power efficiency. Figure 9 shows that for *R* of 3, the dimming range of DPO is 8–92%, which is close to that of HDAP with a 7–93% dimming range. Moreover, DPO provides better *R* for dimming range 40–60% compared to other OFDM techniques discussed here.

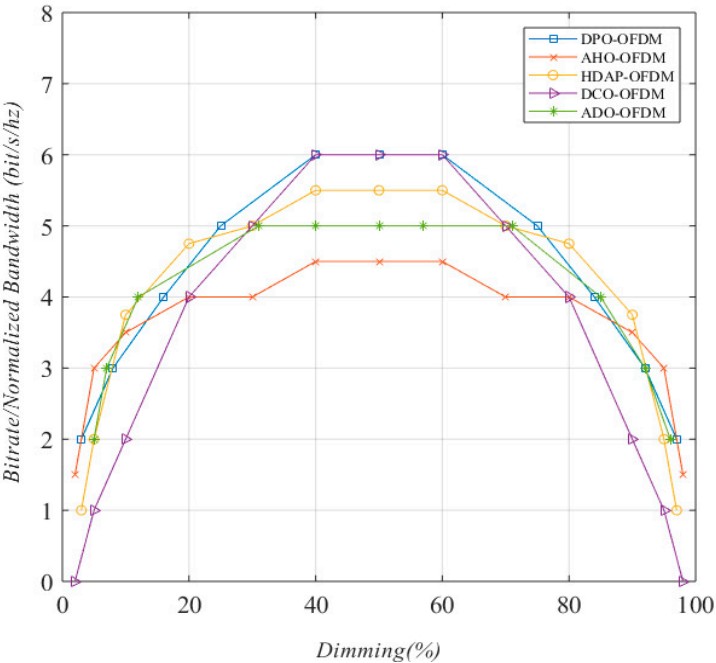

**Figure 9.** Dimming Range of dc-biased pulse amplitude modulated OFDM (DPO-OFDM) and other OFDM forms where OFDM stands for orthogonal frequency division multiplexing.

Next, we show the dimming control of a hybrid OFDM scheme by switching its individual components. For instance, HDAP is considered in evaluating the effectiveness of switching components. Using the concept of AAD-OFDM reported in [24], we can obtain greater spectral efficiency, as well as a wide dimming range with dimming control for the components of HDAP. In HDAP, DCO is placed in lower half subcarriers. In the upper half, odd and even imaginary parts ACO and PAM-DMT are placed, respectively. The upper portion of HDAP, which is a combination of ACO-PAM, acts just like AHO. It can be seen from Figure 9 that for very low dimming values such as 2–10% and very high dimming value 90–98% AHO, it means that ACO-PAM has better spectral efficiency than any other OFDM techniques. Again for 10–30% dimming value and 70–90% dimming value, stand-alone HDAP achieves better spectral efficiency. For mid dimming values such as 30–70%, DCO has greater spectral efficiency. Next, the BER performance is evaluated for different dimming levels. Note that the plots will vary depending on the parameters considered. In this case, we consider the concepts reported in [27]. Figure 10 shows the BER versus $E_{b(elec)}/N_O$ for DPO when the dimming is varied from 10% to 50%. It can be seen that the best BER performance is achieved at 50% dimming. The performance degrades symmetrically when the dimming increases or decreases from 50%. This means dimming at 90% will lead to the same BER performance as of 10%, and dimming at 70% will lead to the same performance as that of 30%. Now, if we switch the OFDM techniques according to our required dimming level, then we can achieve better spectral efficiency with the widest

dimming range using the full dynamic range of LED light for HDAP elements. Applying this concept, a wide dimming range with better spectral efficiency is achieved. Figure 11 shows a block diagram of the transmitter of this combining technique.

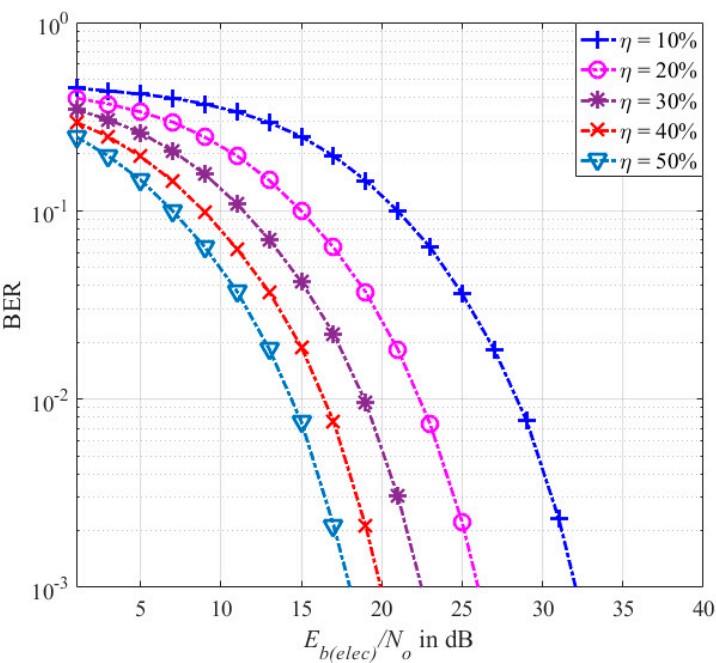

**Figure 10.** Bit error rate (BER) versus electrical energy per bit to noise power spectral density ($E_{b(elec)}/N_O$) for dc-biased pulse amplitude modulated orthogonal frequency division multiplexing (DPO-OFDM) for different dimming levels, $\eta$.

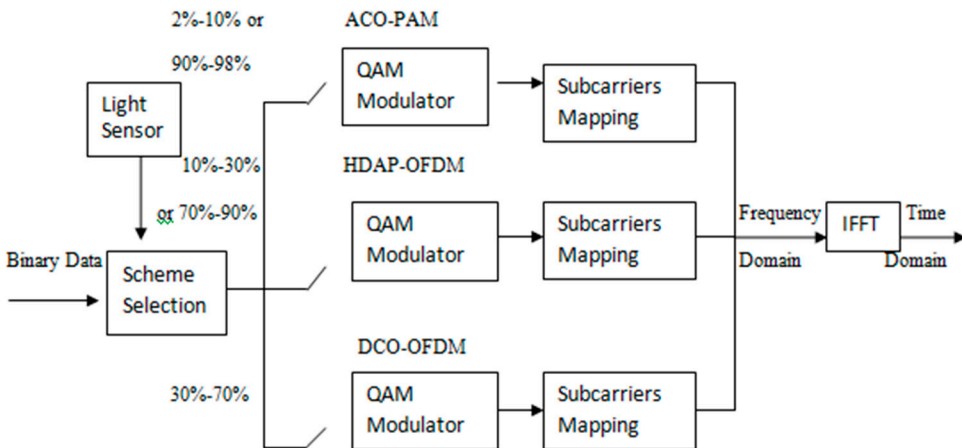

**Figure 11.** Block Diagram of the transmitter part for switching between different components of hybrid DC biased asymmetrically clipped pulse amplitude modulated discrete multitone modulation optical OFDM (HDAP-OFDM), where OFDM stands for orthogonal frequency division multiplexing.

By applying this technique, greater spectral efficiency with a wide dimming range can be achieved using different elements of HDAP and HDAP itself. A light sensor at the start of the transmitter section observes the ambient light and its changes. This light sensor will calculate the dimming level. After obtaining the required dimming level, the scheme selection block will select the suitable scheme. In short, when the brightness is very low or very high, then the ACO-PAM scheme will carry the data. Again, in the lower mid brightness, a stand-alone HDAP scheme will be activated and carry the data. Lastly, when the brightness is in the mid-level, then DCO will carry the data. The term $\eta_r$ is assumed as the required dimming level. The target bit error rate, BER, is $10^{-3}$. Furthermore, $M_{ACO}$,

$M_{DCO}$, $M_{PAM}$ are the sizes of the constellation points for ACO, DCO, and PAM-DMT, respectively. By which spectral efficiency can be achieved, and it is measured in bit/s/Hz. Here, $I_L = 0$ and $I_H = 1$, and the average amplitude of the signal $I_D$. With these assumptions, Algorithm 1. presents the switching method between ACO-PAM, HDAP, and DCO for obtaining wider dimming range and better spectral efficiency.

---

**Algorithm 1** Pseudocode for switching method between ACO-PAM, HDAP, and DCO

---

1: Input $I_L = 0$, $I_H = 1$, BER = $10^{-3}$, $N \geq 128$;
2: Compute the required dimming level $\eta_r$;
3: IF ($\eta_r$ = 2% to 10% or $\eta_r$ = 90% to 98%);
4: THEN select the ACO-PAM scheme;
5: ELSE IF ($\eta_r$ = 10% to 30% or $\eta_r$ = 70% to 90%);
6: THEN select the stand-alone HDAP (50% DCO and 50% ACO-PAM) scheme;
7: ELSE;
8: ·select the DCO scheme.

---

## 7. Application of ML on Hybrid OFDM Forms

As a subset of artificial intelligence, ML was widely used for tasks such as prediction, classification, and detection in various application scenarios [32–38]. Some of these applications are in fields of biomedical engineering [32,38], electrical engineering [33], petroleum engineering [34], computer engineering [35], urban planning engineering [36], and software engineering [37]. ML can also be useful in the context of telecommunication and LiFi. ML regressions are used to predict numerical values based on some system attributes. A regression model can be developed using a set of training samples. This model can produce predicted results. The effectiveness of a regression model can be evaluated using the coefficient of determination or $R^2$ score and root-mean-square error (RMSE). This $R^2$ is a goodness of fit measurement of the models, which indicates how well the predicted values fit or make an approximation of the actual samples [38]. Hence, a unity value of $R^2$ means that the regression model is a perfect fit for the actual samples. On the other hand, RMSE indicates how concentrated the data samples are around the best-fitted line. An RMSE value near zero means there is very low predicted errors, and the model fits the actual data points [38].

This section discusses the application of ML in finding the individual components of hybrid OFDM forms. The methods used in this section can be applied to any hybrid optical OFDM, but we focus our study on DPO. In this case, ML regressors can be used to find the appropriate constellation size of one component of DPO, that is, DCO or PAM-DMT, when the other component is given and when the target dimming level is known. To apply regression algorithms, a dataset was generated by simulating the DPO-based LiFi system using the MATLAB tool. The obtained dataset has 200 sample values, and the samples are randomly placed in the dataset. In the dataset, the input attributes are the subcarrier number $N$, the number of symbols, the constellation size of the PAM-DMT element denoted as $M_{PAM}$, and the dimming level. The output or target attribute of the dataset is the constellation size of the DCO element denoted as $M_{DCO}$. The term $E_{b(elec)} / N_O$ was varied to keep the BER to a constant value of $10^{-3}$. Python programing language was used for analyzing the data and generate the plots. For that, the Anaconda distribution package, Scikit-learn library, Jupiter notebook, and Spyder (version 3.3.3) were used to simplify package management and deployment of Python. Different regression algorithms, including linear regression and polynomial regression, were applied. In regressions, the output depends on single or multiple independent variables. The general equation for multiple variable linear regression is as follows.

$$y = m_0 + m_1 x_1 + m_2 x_2 + \ldots + m_n x_n \tag{15}$$

In (15), $x_1, x_2, \ldots, x_n$ are the input variables, and $m_0, m_1, \ldots, m_n$ are the estimators of the regression coefficient. On the other hand, polynomial regressions can be expressed in the following format.

$$y = m_0 + m_{11}x_1 + m_{21}x_2 + \ldots + m_{n1}x_n + m_{12}x_1{}^2 + m_{22}x_2{}^2 + \ldots + m_{n2}x_n{}^2 + \ldots + m_{nn}x_n{}^n \quad (16)$$

In (16), $m_0, m_{11}, m_{21} \ldots m_{n1}, m_{12}, m_{22}, m_{n2}, m_{nn}$ are the estimators of the regression coefficient, and $n$ is the degree of the polynomial. In the case of our experiments with polynomial regression, the order was varied from 2 to 5. The data were split into training and testing samples. The percentage of testing and training samples were varied. Investigations were conducted for a testing size of 10%, 20%, and 30% of the total samples, where the remaining samples were used for training the model. The regression coefficients were estimated using least square algorithms. Figure 12 shows the predicted value versus the actual value of the DCO constellation size for the case of linear regression. It can be seen that the plots do not follow the linear relationship in most cases. Figure 13 shows the predicted value versus the actual value of the DCO constellation size for the case of polynomial regression of order 3. The plots of Figure 13 show an approximately linear relationship. The effectiveness of the plots is calculated using the coefficient of determination or $R^2$ score, which measures how the predicted values fit the actual values. Table 1 shows the $R^2$ and RMSE values of the linear and polynomial regressions. In Table 1, results are shown for linear regression and polynomial regression of degrees 2, 3, and 4. The results for degree 5 and above are omitted as the $R^2$ values were very low. Table 1 shows that linear regression has a low $R^2$ value and thus is not suitable in this case. On the other hand, polynomial regression with degrees 2, 3, and 4 have high $R^2$ scores at different values of testing and training samples. This high $R^2$ score, as described in the literature [38], can ensure the fitness of the model. It can be seen from Table 1 that the best performance can be obtained when polynomial regression of order 4 was applied to estimate the constellation size of the DCO component of DPO for a given dimming target and given constellation size of PAM-DMT. This is for the case of 10% testing and 90% training samples. The high $R^2$ score of 98.18% and low RMSE value of 1.18 indicate that polynomial regression of order 4 is suitable [38] for predicting DCO elements within DPO-based LiFi. Similarly, the constellation size of the PAM-DMT component of DPO can be estimated for a given dimming target and given constellation size of DCO.

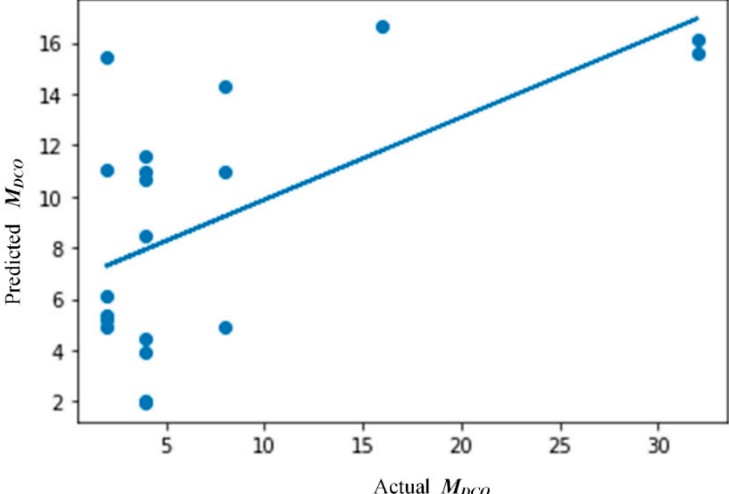

**Figure 12.** Predicted versus actual $M_{DCO}$ of dc-biased pulse amplitude modulated orthogonal frequency division multiplexing (DPO-OFDM) for the case of linear regression. Note that $M_{DCO}$ stands for the constellation size of DC-biased optical OFDM (DCO-OFDM).

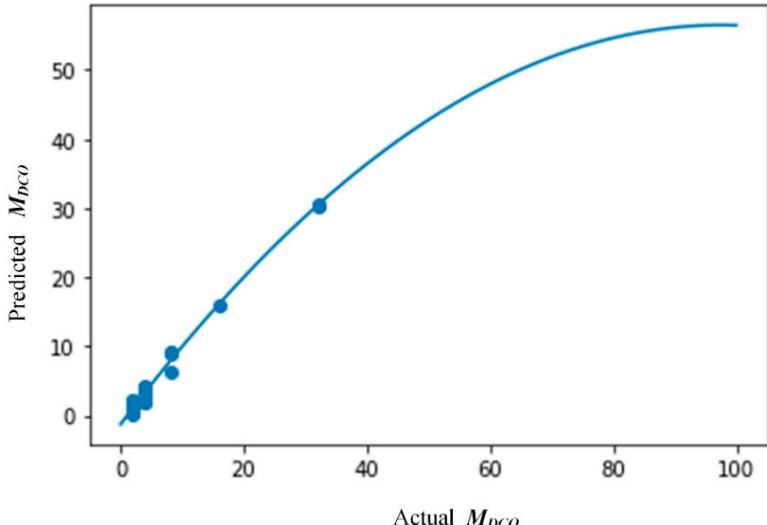

**Figure 13.** Predicted versus actual $M_{DCO}$ of dc-biased pulse amplitude modulated orthogonal frequency division multiplexing (DPO-OFDM) for the case of polynomial regression of degree = 4. Note that $M_{DCO}$ stands for the constellation size of DC-biased optical OFDM (DCO-OFDM).

**Table 1.** Performance of different regression algorithms.

| Regression Algorithm | Testing% | $R^2$ Score | RMSE |
|---|---|---|---|
| Linear regression | 10% | 31.30% | 7.30 |
| Polynomial regression degree 2 | 10% | 87.34% | 3.13 |
| Polynomial regression degree 3 | 10% | 97.17% | 1.48 |
| Polynomial regression degree 4 | 10% | 98.18% | 1.18 |
| Polynomial regression degree 5 | 10% | 30.99% | 7.31 |
| Polynomial regression degree 2 | 20% | 89.85% | 3.07 |
| Polynomial regression degree 3 | 20% | 95.72% | 1.99 |
| Polynomial regression degree 4 | 20% | 97.53% | 1.51 |
| Polynomial regression degree 2 | 30% | 86.77% | 3.39 |
| Polynomial regression degree 3 | 30% | 93.90% | 2.30 |
| Polynomial regression degree 4 | 30% | 95.26% | 2.02 |

## 8. Conclusions

In this work, a new OFDM format, DPO scheme, is proposed. In DPO, the two components DCO and PAM-DMT are carried by the odd subcarriers and the imaginary part of the even subcarriers, respectively. A mathematical expression for the DC-bias level is derived considering a target dimming level for DPO. The simulation results show that to reach a BER of $10^{-3}$, DPO requires almost 2 dB less electrical power than HDAP, as well as 7 dB less electrical power than AHO. Moreover, DPO has great spectral efficiency and a wider dimming range of 8–92%, close to HDAP. Therefore, it provides better performance than HDAP and AHO in terms of electrical power and spectral efficiency. Moreover, this paper introduces a switching scheme for HDAP where the individual components of HDAP are switched in obtaining a wide dimming range at an excellent spectral efficiency. Finally, this paper applies ML regression algorithms to find the constellation sizes of DPO elements. It is shown here that for a given target dimming level, polynomial regression of order 4 can successfully determine the required constellation size of one element of DPO if the other constellation size of the other element is known.

In the future, ML algorithms can also be used to predict the LiFi channels. Furthermore, the performance of DPO can be evaluated considering other channel impairments, including co-channel interference and side effect modulation.

**Supplementary Materials:** The following are available online at https://www.mdpi.com/article/10.3390/electronics10172103/s1, Table S1: Dataset for DPO-OFDM.

**Author Contributions:** Conceptualization, M.R.H.M.; methodology, I.N., M.R.H.M. and R.I.; software, I.N. and R.I.; validation, I.N. and R.I.; formal analysis, I.N., M.R.H.M. and R.I.; investigation, I.N., M.R.H.M. and R.I.; data curation, I.N., M.R.H.M. and R.I.; writing—original draft preparation, I.N.; writing—review and editing, M.R.H.M. and J.K.; visualization, I.N. and R.I.; supervision, J.K.; project administration, J.K.; funding acquisition, J.K. All authors have read and agreed to the published version of the manuscript.

**Funding:** This research received no external funding. The APC will be funded by the School of Engineering, IT and Physical Sciences, Federation University Australia.

**Data Availability Statement:** The dataset generated in this study is uploaded as Supplementary Material.

**Conflicts of Interest:** The authors declare no conflict of interest.

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
