# Peer review of "A Novel OFDM Format and a Machine Learning Based Dimming Control for LiFi"

_electronics, doi:10.3390/electronics10172103_

Round 1

Reviewer 1 Report

The authors present an hybrid strategy for loading M-QAM and PAM signals in OFDM for LiFi taking into account dimming constraints. Results show that this strategy can obtain a larger spectral efficiency for the same Eb/No when compared to current OFDM waveforms for optical wireless communications.

My suggestion is that define formulas that you are plotting since they are not defined, plot BER for different dimming levels (e.g. low medium and high) and evaluate test the spectral efficiency in terms of the for a low, medium and high Eb/No. Test for a Eb/No of 40 dB is not realistic and you are losing part of the photo.  Also describe the optimization algorithm of the dimming using Machine Learning because it is not said in any place what it is the function that optimizes and how.

Reviewer 2 Report

Reviewer’s Recommendation:

This is nice work and it shows good findings. Nevertheless, some major revisions are needed for enhancing and clarification purposes.

Summary

This manuscript presents an OFDM scheme with a machine learning (ML) intended for LiFi.

General comments

This manuscript has something important to show. The authors have worked towards introducing a viable scheme for LiFi. There are still some revisions that should be conducted. The mathematical aspect should be enhanced in order to present a model if possible. Also, some figures are missing (please see below) and a better expression is needed in some cases. In general, this candidate paper is eligible for publication as soon as all the revisions are accomplished.

Suggested Improvements

In conjunction to the aforementioned, please apply the suggested corrections inside the manuscript with different color:

  1. In abstract, you should elaborate a little more in the end highlighting the advantages of your work. It will help the diffusion.
  2. Relevant to OFDM please see "Preliminary BER Study of a TC-OFDM system operating under noisy conditions" and "Performance of Turbo Coded OFDM Under the Presence of Various Noise Types".
  3. You mention that "A new modulation scheme termed as DPO-OFDM is introduced...". If you mean OFDM, please be informed that OFDM is not a modulation but rather a multiplexing technique. Please avoid these kind of confusions for the safety of your work.
  4. In 2. LITERATURE REVIEW ON OFDM FORMATS, you should declare that this a review without rules (e.g. SALSA, PRISMA, etc.) but please present the utilized databases and the keywords for finding such a bibliography.
  5. "The performance results of DPO and other OFDM schemes are obtained here with the help of simulations with MATLAB tool". Please refer to the version of MATLAB/Simulink.
  6. Relevant to Figures 4,5 you should add also BER curves (Signal power versus bit-error-rate).
  7. You use OFDM in Li-Fi. A further analysis of various numbers subcarriers is needed in order to show or not the impact in such kind of communications.

Manuscript Rating:

  1. In this candidate paper, significant work has been conducted but still are some revisions to be done. Please apply them.

Reviewer 3 Report

This paper proposes a new hybrid orthogonal frequency division multiplexing (OFDM) form termed as DC-biased pulse amplitude modulated optical OFDM (DPO-OFDM) by combining the ideas of existing DC-biased optical OFDM (DCO-OFDM) and pulse amplitude modulated discrete multitone (PAM-DMT).

The paper’s subject could be interesting for readers of journal. Therefore, I recommend this paper for publication in this journal but before that, I have a few comments on the text that should be addressed before publication:

Comments:

1)In page 4: Equation (5) and Equation (4) are not clear. They are one side equations!. Authors should clarify that.

2)In page 6 Figure 2: The title of figure (2) is too long. The title of figures should be shorter. Authors can explain more about the figure after or before that, not in the title.

3)In page 13 Figure 9: Font of the vertical and horizontal axis of the chart is too big. Authors should make it smaller than what it is now.

4) In page 14, Patent section: The statement in this section is a template. Authors should remove that or fill it with their own sentences.

5)About accuracy of the presented model, authors have used RMSE and R criteria. The problem is that there is no obvious interpretation of obtained values of RMSE and R. In other words, authors should explain why the obtained values are acceptable. Are they acceptable in comparison with previous similar works and articles?

6)In the Abstract section: The novelty of conducted research by authors should be bolder. In other words, what is new in this article in comparison with similar works?

7)Which softwares have been used for modelling the data and also applying the Machine learning Algorithms?

8)Which softwares have been used to draw and export the charts?

9)How many percent of the data have been used as training, validating and testing data by the authors in this work?

10) Since recently it has been proved that machine learning and artificial intelligence (AI) has numerous applications in all of engineering fields, I highly recommend the authors to add some references in this manuscript in this regard. It would be useful for the readers of journal to get familiar with the application of AI in other engineering fields. I recommend the others to add all the following references, which are the newest references in this field of biomedical engineering [1], electrical engineering [2], petroleum engineering [3], computer engineering [4], and urban planning engineering [5], software engineering

[1] Tavakoli, S., & Yooseph, S. (2019). Learning a mixture of microbial networks using minorization–maximization. Bioinformatics, 35(14), i23-i30.

[2] Jahanshahi, A. (2019). TinyCNN: A Tiny Modular CNN Accelerator for Embedded FPGA. arXiv preprint arXiv:1911.06777.

[3] Roshani M, Proposing a gamma radiation based intelligent system for simultaneous analyzing and detecting type and amount of petroleum by-products, Nuclear Engineering and Technology 53 (4), 1277-1283.

[4] Voghoei, S., Tonekaboni, N. H., Yazdansepas, D., & Arabnia, H. R. (2019, December). University Online Courses: Correlation between Students' Participation Rate and Academic Performance. In 2019 International Conference on Computational Science and Computational Intelligence (CSCI) (pp. 772-777). IEEE..

[5] Arabi, M., Beheshtitabar, E., Ghadirifaraz, B. and Forjanizadeh, B., 2015. Optimum Locations for Intercity Bus Terminals with the AHP Approach–Case Study of the City of Esfahan. World Academy of Science, Engineering Technology, International Journal of Social, Behavioral, Educational, Economic, Business Industrial Engineering, 9(2), pp.545-551.

[6] Chenarlogh, V. A., Razzazi, F., & Mohammadyahya, N. (2019, December). A Multi-View Human Action Recognition System in Limited Data Case using Multi-Stream CNN. In 2019 5th Iranian Conference on Signal Processing and Intelligent Systems (ICSPIS) (pp. 1-11). IEEE.

Round 2

Reviewer 1 Report

The authors have done my comments. So, the paper could be published in its current format. 

Author Response

We are glad that the reviewer was satisfied with the changes we made in the previous revision.

We thank the reviewer for recommending acceptance of the paper.

Reviewer 2 Report

The authors have made successfully the needed revisions. Still, there are some minor revisions to be done. No other review is needed for speeding the process. The handling editor can supervise it. Please see:

  • Relevant to my comment that “you use OFDM in Li-Fi. A further analysis of various numbers subcarriers is needed in order to show or not the impact in such kind of communications”, you made revisions. Please present the relevant figures of the aforementioned inside manuscript.
  • A final grammatical and expressional check should be made.

Sincerely,

The reviewer

Reviewer 3 Report

All the comments have been addressed correctly. The paper is ready for publication in the present form. 

Author Response

(The authors gave the same response as above.)
